# Citrullination profile analysis reveals peptidylarginine deaminase 3 as an HSV-1 target to dampen the activity of candidate antiviral restriction factors

**Selina Pasquero**[1]☯, **Francesca Gugliesi**[1]☯, **Matteo Biolatti**[1], **Valentina Dell'Oste**[1], **Camilla Albano**[1], **Greta Bajetto**[1,2], **Gloria Griffante**[3], **Linda Trifirò**[1], **Bianca Brugo**[1], **Stefano Raviola**[2,3], **Davide Lacarbonara**[2,3], **Qiao Yang**[1,4], **Sen Sudeshna**[5], **Leonard Barasa**[5], **Hafeez Haniff**[5], **Paul R. Thompson**[5], **Santo Landolfo**[1], **Marco De Andrea**[1,2]*

1 Department of Public Health and Pediatric Sciences, University of Turin – Medical School, Turin, Italy,
2 CAAD Center for Translational Research on Autoimmune and Allergic Disease, University of Piemonte Orientale, Novara Medical School, Novara, Italy, 3 Department of Translational Medicine, University of Piemonte Orientale, Novara, Italy, 4 Avian Disease Research Center, College of Veterinary Medicine, Sichuan Agricultural University, Wenjiang, Chengdu City, P.R. China, 5 Department of Biochemistry and Molecular Pharmacology, UMass Medical School, Worcester, Massachusetts, United States of America

☯ These authors contributed equally to this work.
* marco.deandrea@unito.it

**Data Availability Statement:** All relevant data are within the manuscript and its Supporting information files.

## Abstract

Herpes simplex virus 1 (HSV-1) is a neurotropic virus that remains latent in neuronal cell bodies but reactivates throughout an individual's life, causing severe adverse reactions, such as herpes simplex encephalitis (HSE). Recently, it has also been implicated in the etiology of Alzheimer's disease (AD). The absence of an effective vaccine and the emergence of numerous drug-resistant variants have called for the development of new antiviral agents that can tackle HSV-1 infection. Host-targeting antivirals (HTAs) have recently emerged as promising antiviral compounds that act on host-cell factors essential for viral replication. Here we show that a new class of HTAs targeting peptidylarginine deiminases (PADs), a family of calcium-dependent enzymes catalyzing protein citrullination, exhibits a marked inhibitory activity against HSV-1. Furthermore, we show that HSV-1 infection leads to enhanced protein citrullination through transcriptional activation of three PAD isoforms: PAD2, PAD3, and PAD4. Interestingly, PAD3-depletion by specific drugs or siRNAs dramatically inhibits HSV-1 replication. Finally, an analysis of the citrullinome reveals significant changes in the deimination levels of both cellular and viral proteins, with the interferon (IFN)-inducible proteins IFIT1 and IFIT2 being among the most heavily deiminated ones. As genetic depletion of IFIT1 and IFIT2 strongly enhances HSV-1 growth, we propose that viral-induced citrullination of IFIT1 and 2 is a highly efficient HSV-1 evasion mechanism from host antiviral resistance. Overall, our findings point to a crucial role of citrullination in subverting cellular responses to viral infection and demonstrate that PAD inhibitors efficiently suppress HSV-1 infection *in vitro*, which may provide the rationale for their repurposing as HSV-1 antiviral drugs.

**Funding:** This research was supported by the University of Turin (RILO2021 and RILO2022 to M. D.A., F.G., V.D.O. and M.B.), by the Ministry of Education, University and Research – MUR (PRIN Project 2017ALPCM to V.D.O.), by Cassa di Risparmio of Turin Foundation (RF=2021.1745 to M.B.), and by EU funding within the MUR PNRR Extended Partnership initiative on Emerging Infectious Diseases (Project no. PE00000007, INF-ACT to M.D.A). The funders had no role in study design, data collection and analysis, decision to publish, or preparation of the manuscript.

**Competing interests:** The authors have declared that no competing interests exist.

## Author summary

HSV-1 is a common human pathogen that infects approximately 70% of the population for life. After infection, the virus remains dormant in sensory neurons but reactivates periodically, releasing virus particles that move down the axon to infect skin epithelial cells, where it can be spread to other individuals. Depending on the recipient's immune condition, primary infection or reactivation can cause a wide range of symptoms. However, accessible HSV-1 antivirals are currently limited, and most of them target viral DNA polymerase, thus leading to the emergence of drug-resistant viral infections. Citrullination, an irreversible protein alteration driven by peptidylarginine deiminases (PADs), has been linked to various inflammation-related events, including viral infections. In our study, we provide evidence that HSV-1 triggers citrullination of multiple proteins, some of which possess anti-viral properties, thereby promoting viral fitness. Notably, we show that specifically targeting the PAD3 isoform dramatically reduces viral replication. Overall, our study sheds light on the potential of using host PAD inhibitors for developing antiviral agents against HSV-1.

## Introduction

Herpes simplex virus 1 (HSV-1) is a widespread and highly infectious alpha-herpesvirus, with seroprevalence reaching up to 75% in the adult population [1]. Primarily transmitted by oral-oral contact and often causing orolabial herpes, commonly known as "cold sores", HSV-1 can persist latently and lifelong in the local ganglia of its host. In response to a variety of diverse stimuli, the virus can reactivate to produce new virus progeny. Reactivation results in clinical signs and symptoms ranging from painful, but self-limited, infections of the oral or genital mucosa to severe infections of the eye or life-threatening infections in immuno-compromised hosts or newborns [2]. Additionally, recent evidence suggests that HSV-1 may be involved in the etiology of Alzheimer's disease (AD) [3,4]. Although numerous vaccine candidates have been investigated in clinical trials, no licensed vaccine is available to prevent HSV infections [5].

Acute HSV infections can be treated with antiviral drugs that inhibit its DNA-polymerase. Even though these drugs have been approved and used for decades, their effectiveness has been hampered by the emergence of drug resistant HSV strains [6]. A new strategy to develop antiviral drugs that can overcome these obstacles is to target the host cell factors that partici-pate in viral replication. These drugs are known as host-targeting antivirals (HATs) [7,8].

An example of a host cell factor that can be targeted by HATs is peptidylarginine deimi-nases (PADs). PADs are a family of enzymes that need calcium to work and can change the structure of proteins by a process called citrullination. This post-translational modification (PTM), also called deimination, is a process in which the guanidinium group of a peptidyl-arginine is hydrolyzed to form peptidyl-citrulline, a non-genetically coded amino acid [9,10]. Five PAD isozymes (PADs 1–4 and 6) are expressed in humans, with a unique distribution in various tissues [9,11,12], and their dysregulation has been associated with various inflamma-tory conditions and neurodegenerative disorders, such as multiple sclerosis (MS) and AD [13–18]. Given the involvement of PADs in several pathological settings, several PAD inhibitors have been synthesized in recent years. Some of these compounds, such as Cl-amidine (Cl-A) and its derivative BB-Cl-amidine (BB-Cl) [19–21], can inhibit the activity of all the different isoforms and, as such, are called pan-PAD inhibitors [19,22]. Other available inhibitors are

highly specific for the different PAD-isozymes, like AFM-30 for PAD2, CAY10727, and HF4 for PAD3, and GSK199 for PAD4 [23,24].

There is growing evidence of a link between PAD dysregulation and viral infections. Specifically, Arisan and colleagues [25] have recently shown that SARS-CoV-2 infection can modulate PADI gene expression, especially in lung tissues. In addition, we have recently demonstrated an *in vitro* antiviral activity of PAD inhibitors against betacoronaviruses [26]. It has also been found that the antiviral activity of the LL37 protein is weakened by human rhinovirus-induced citrullination [27], and that sera from rheumatoid arthritis (RA) patients specifically recognize artificially citrullinated Epstein-Barr virus proteins [28,29]. Finally, we have demonstrated that another member of the *Herpesviridae* family, human cytomegalovirus (HCMV), upregulates two members of the PAD family (*i.e.*, PAD2 and PAD4) to promote viral fitness, while PAD-inhibitors significantly dampen HCMV replication *in vitro* [30].

Here we report that HSV-1 can alter the protein citrullination profile in different cell lines upon infection, and that this function is mediated by the specific activation of PAD3. Furthermore, by analyzing the citrullinome profile, we also identify IFIT1 and IFIT2 as potential restriction factors for HSV-1 replication. Our findings suggest that: i) HSV-1 replication can be blocked by a new type of HTAs, namely PAD inhibitors, with PAD3 and its specific inhibitor being key players, and ii) HSV-1 manipulates IFIT1/2 to enhance its replication rate.

The potential impact of peptidylarginine deiminases as novel targets for antiviral therapy against HSV-1 are further discussed below.

## Results

### Pan-PAD inhibitors block HSV-1 replication

We have previously shown that HCMV triggers PAD-mediated citrullination to enhance its replication [30]. To test whether this feature would apply to other herpesviruses, we first measured total protein citrullination in human foreskin fibroblasts (HFFs) upon HSV-1 infection (MOI 1) at different time points. We used electrophoresis to analyze the protein lysates incubated with the citrulline-specific probe Rh-PG. HSV-1-infection increased and altered the pattern of total protein citrullination in lysates from infected *vs* uninfected mock cells, starting from 16 h post-infection (hpi) (Fig 1A, left panel). In particular, we observed a specific enrichment of bands corresponding to proteins with molecular weights higher than 75 kDa, whereas the total amount of proteins remained unchanged, as shown by the similar blue Coomassie staining at different time points (Fig 1A, right panel).

Since citrullination is catalyzed by the PAD family of enzymes [31], we tested whether the cell-permeable pan-PAD inhibitors Cl-amidine (Cl-A) and BB-Cl-amidine (BB-Cl) would affect HSV-1 replication. For this purpose, we assessed viral plaque formation in HSV-1-infected HFFs (MOI 1) treated for 1 h before infection with increasing concentrations of Cl-A (25–200 μM; Fig 1B) or BB-Cl (0.6–5 μM; Fig 1C). After 24 h of continuous exposure to the inhibitors, we observed a dose-dependent decrease in the number of viral particles in all treated cells. The IC50 values of Cl-A and BB-Cl were ~61 μM and ~0.7 μM, respectively, which is consistent with our previous results [30]. Notably, 100 μM Cl-A and 2.5 μM BB-Cl were able to significantly reduce the virus yield by approximately 1 log. We also assessed cell viability using the 3-(4,5-dimethylthiazol-2-yl)-2,5-diphenyl tetrazolium bromide (MTT) assay in cells treated under the same conditions and found that the drugs were not cytotoxic at these concentrations (S1A and S1B Fig). To confirm these results in other cell types, we repeated the experiments in adult retinal pigment epithelial (ARPE-19) and neuroblastoma (SH-SY5Y) cells. We observed similar antiviral effects of Cl-A in both cell lines (S1C and S1D Fig), with only a slight cytotoxic effect in SH-SY5Y cells at higher concentrations (S1A Fig). These

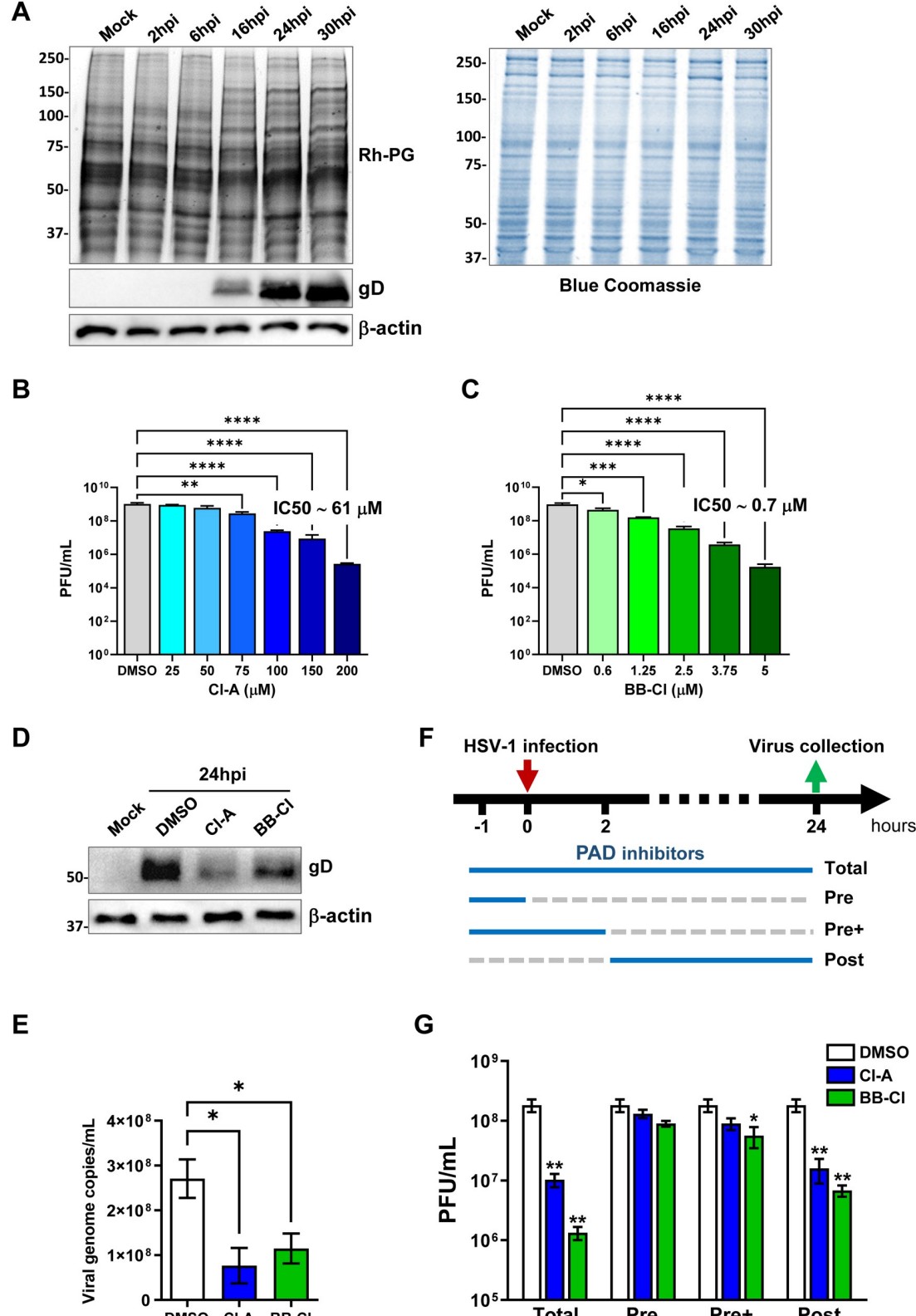

**Fig 1. The pan-PAD inhibitors Cl-A and BB-Cl inhibit HSV-1 replication in human fibroblasts. (A)** Protein lysates from HFFs infected with HSV-1 (MOI 1 PFU/cell) at different hours post-infection (hpi) or from uninfected HFFs (Mock) were exposed to an Rh-PG citrulline-specific probe (left panel) and subjected to gel electrophoresis to detect citrullinated proteins. An

anti-gD antibody was used to assess HSV-1 infection, while β-actin cellular expression was used for protein loading control. Equal loading was also assessed by Coomassie blue staining (right panel). One representative gel of three independent experiments is shown. (**B, C**) Dose-response curves of the cell-permeable pan-PAD inhibitors Cl-A (B) and BB-Cl (C) in HFFs infected with HSV-1 (MOI 1). Inhibitors were given 1 h prior to virus adsorption and kept throughout the whole experiment. 24 hpi viral plaques were microscopically counted and the number of plaques was plotted as a function of inhibitor concentration. Values are expressed as mean ± SEM of three independent experiments, $^*P < 0.05$, $^{**}P < 0.01$, $^{***}P < 0.001$; one-way ANOVA followed by Bonferroni's post test. (**D**) Protein lysates from uninfected (mock) or infected HFFs (24 hpi) at an MOI of 1 PFU/cell treated with Cl-A (100 μM), BB-Cl (2.5 μM), or vehicle (DMSO) were analyzed by immunoblotting for viral expression (gD) and normalized to β-actin. (**E**) To determine the number of viral DNA genomes in HSV-1-infected HFFs, viral DNA was isolated at 24 hpi and analyzed by qPCR using primers amplifying a region of the gE gene. GAPDH was used to normalize HSV-1 genome counts. Values are expressed as mean ± SEM of three independent experiments. HFFs were infected with HSV-1 (MOI 1 PFU/cell) and then treated with Cl-A (100 μM), BB-Cl (2.5 μM), or vehicle (DMSO), which were given at four different time points as indicated (**F**). At 24 hpi, viral plaques were microscopically counted and expressed as PFU/mL (**G**). Values are expressed as mean ± SEM of three independent experiments, $^*P < 0.05$, $^{**}P < 0.01$; one-way ANOVA followed by Bonferroni's post test.

findings suggest that citrullination is a common and important event for HSV-1 replication in different cell models.

We next examined the effect of Cl-A and BB-Cl on HSV-1 protein expression and citrullination in HFFs. We treated HSV-infected cells with the same concentrations of the drugs as before and collected them at 24 hpi. We found that both drugs significantly reduced the expression of gD, a late viral protein, compared to the vehicle control (Fig 1D). We also observed that the drugs restored the normal citrullination profile of the cells, which was altered by the infection (S1E Fig). Moreover, the drugs prevented the cytopathic effect caused by the virus (S1F Fig).

To measure viral DNA replication, we performed qPCR analysis on the same cell samples. We detected a slight decrease in viral DNA copies in the drug-treated cells, albeit not as pronounced as that observed in the plaque assay (Fig 1E), suggesting that PAD inhibitors may target multiple stages of the HSV-1 life cycle.

To further investigate this possibility, we tested tree different treatment schedules using Cl-A and BB-Cl (Fig 1F). We either added the drugs before infection and washed them away prior to viral addition (pre-treatment, Pre) or removed them at 2 hpi (pre-treatment + infection, Pre+). We observed only a slight difference in the number of viral particles counted by plaque assay between Pre and Pre+ cells, with neither group achieving a notable reduction in the number of viral particles when compared to Post cells (Fig 1G).

These data suggest that PAD activity plays a less important role in the early phases of infection, such as binding and entry, but becomes essential during the later stages of viral replication.

## HSV-1 infection upregulates PAD expression

To determine whether HSV-1-enhanced citrullination was modulated by any of the five known PAD isoforms (*i.e.*, PADs 1–4 and PAD6) and if they played a role in this process, we conducted RT-qPCR analysis in infected HFFs. The results revealed that the *PADI2*, *3*, and *4* genes were all expressed at significantly higher levels in HSV-1-infected HFFs at 16 hpi compared to mock-infected controls (Fig 2A). By contrast, the other PAD isoforms (*PADI1* and *6*) were expressed at very low levels and did not significantly change following HSV-1 infection (S2A Fig). Consistent with their mRNA expression, PAD2, 3, and 4 protein levels were increased upon HSV-1 infection, albeit with different kinetics (Fig 2B). Compared to uninfected cells, HSV-1-infected HFFs showed a significant increase in PAD2 and PAD4 protein expression (at 16 and 24 hpi, respectively) but returned to basal conditions soon after (S2B Fig). Of note, PAD3 protein was undetectable in mock-infected HFFs, but it was strongly

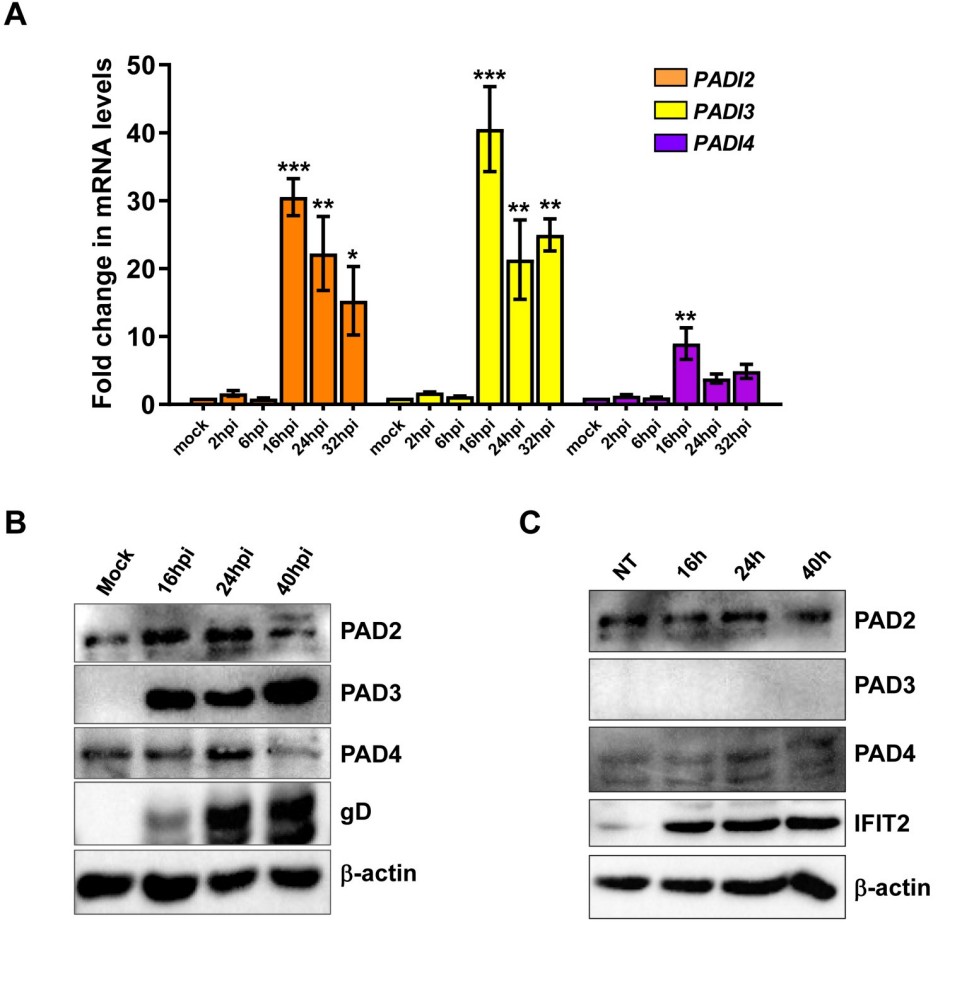

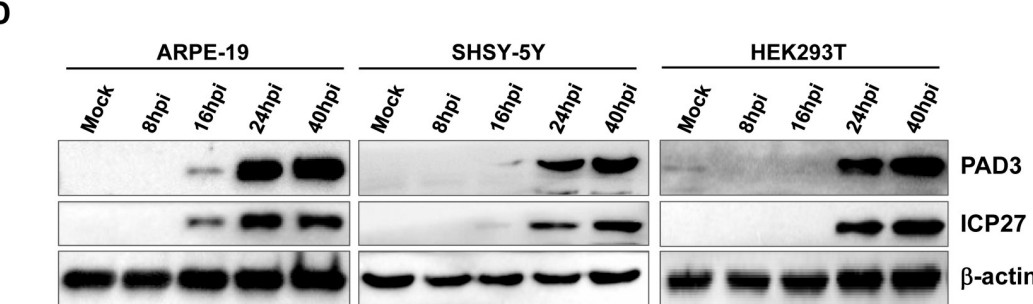

**Fig 2. HSV-1 infection upregulates PAD expression in human fibroblasts. (A)** mRNA expression levels of *PADI* isoforms by RT-qPCR of HSV-1-infected (8 and 16 hpi) *vs* uninfected (mock) HFFs were normalized to the housekeeping gene glyceraldehyde-3-phosphate dehydrogenase (GAPDH) and expressed as mean fold change ± SEM over mock-infected cells. *$P < 0.05$, **$P < 0.01$, ***$P < 0.001$; one-way ANOVA followed by Bonferroni's post test. (**B**) Western blot analysis of protein lysates from uninfected (mock) or infected HFFs (MOI 1) at different time points using antibodies against PAD2, PAD3, and PAD4. An anti-gD antibody was used to assess HSV-1 infection, while β-actin cellular expression was used for protein loading control. (**C**) Western blot analysis of protein lysates from untreated (NT) or IFN-β-treated (500 U/mL) HFFs using antibodies against PAD2, PAD3, and PAD4. An anti-ICP27 antibody was used to assess HSV-1 infection, while β-actin cellular expression was used for protein loading control. (**D**) Western blot analysis of protein lysates from uninfected (mock) or HSV-1-infected (MOI 1) cells at different time points using antibody against PAD3. An anti-ICP27 antibody was used to assess HSV-1 infection, while β-actin cellular expression was used for protein loading control. Representative blots of three independent experiments are shown.

expressed in HSV-1-infected cells starting from 16 hpi and persisted at later time points, with a pattern similar to that of the late gD viral protein. Moreover, to rule out the possibility that increased PAD expression in HFFs was induced by an interferon (IFN) response to viral infection rather than HSV-1 infection *per se*, cells were treated with IFN-β (500 U/mL) and harvested at different time points after treatment. IFN-β treatment failed to change the basal expression of PAD proteins at any time compared to untreated cells, whereas the interferon-inducible IFIT2 protein showed a significant increase (Figs 2C and S2E).

Finally, to ascertain whether this phenomenon was restricted to HFFs, we extended our RT-qPCR and Western blot analyses of PAD3 to include ARPE-19 and SHSY-5Y cells. HEK293 cells, which are known to basally express PAD3 [32], were analyzed as well. In all the cell lines tested, PAD3 was upregulated during HSV-1 infection (MOI 1), with minor variations between the cell lines (Fig 2D and S2C Fig). This pattern contrasts with our earlier findings in HCMV-infected HFFs [30] and HCoV-43-infected MRC-5 fibroblasts [26], where PAD3 was undetectable under basal conditions and did not show any signs of upregulation, suggesting that different viruses may induce different PAD isoforms.

## PAD3 protein levels are induced through a calcium-independent HSV-1-early mechanism

Having established that PAD3 was the most significantly impacted PAD family member by HSV-1 infection, we sought to determine the mechanism underlying *PADI3* transcriptional upregulation in response to viral infection. To this end, we assessed the promoter activity of *PADI3* gene by transiently transfecting HFFs with luciferase reporter plasmids carrying the wild-type promoter region of *PADI3*. At 24 h post transfection, cells were infected with HSV-1. As shown in Fig 3A, HSV-1 infection led to a robust induction of the luciferase activity driven by *PADI3* promoter (~6 fold) and upregulated viral ICP27 expression at 24 hpi (Fig 3B), indicating once more than early stages of infection are critical for the transcriptional activation of *PADI3* gene. As expected, UV-inactivated HSV-1 infection failed to induce both PAD3 and ICP27 protein expression in HFFs compared to cells infected with wild-type HSV-1 (Fig 3B). Furthermore, treatment of HSV-1-infected HFFs with the protein synthesis inhibitor CHX curbed the induction of PAD2, 3, and 4, as well as ICP27, protein expression and total levels of protein citrullination at 24 hpi, without altering total protein levels (Fig 3C and S2E Fig), indicating that *de novo* gene expression is required for PAD protein upregulation and citrullination profile modification during infection. In contrast, treatment with the viral DNA synthesis inhibitor phosphonoformic acid (PFA) only marginally affected the upregulation of PAD3 despite resulting in a dramatic reduction in the synthesis of the late viral protein gD (Fig 3D). Altogether, these data indicate that one or more viral proteins synthetized during the initial stages of infection are involved in PAD3 transcriptional upregulation.

Finally, as HSV-1 infection is known to increase intracellular calcium levels [33], we asked whether PAD3 overexpression could be ascribable to this calcium influx rather than being a direct consequence of upregulated gene expression induced by HSV-1 viral proteins. To rule out this possibility, we employed thapsigargin (TG), an endoplasmic reticulum (ER) stressor known to elevate intracellular calcium levels by interfering with the sarcoplasmic/endoplasmic reticulum $Ca^{2+}$-ATPase [34]. Specifically, we treated mock-infected HFFs with two concentrations of TG (1.5 and 5 μM) and evaluated PAD3 expression at both the mRNA (Fig 3E) and protein (Fig 3F) levels after 16 h. We found that, in comparison with HSV-1-infected cells (MOI 1; 16 hpi), TG treatment failed to upregulate PAD3 expression at any of the concentrations tested, whereas it significantly upregulated *ATF-6* gene expression (Fig 3F), confirming its efficacy as an ER stressor at both concentrations.

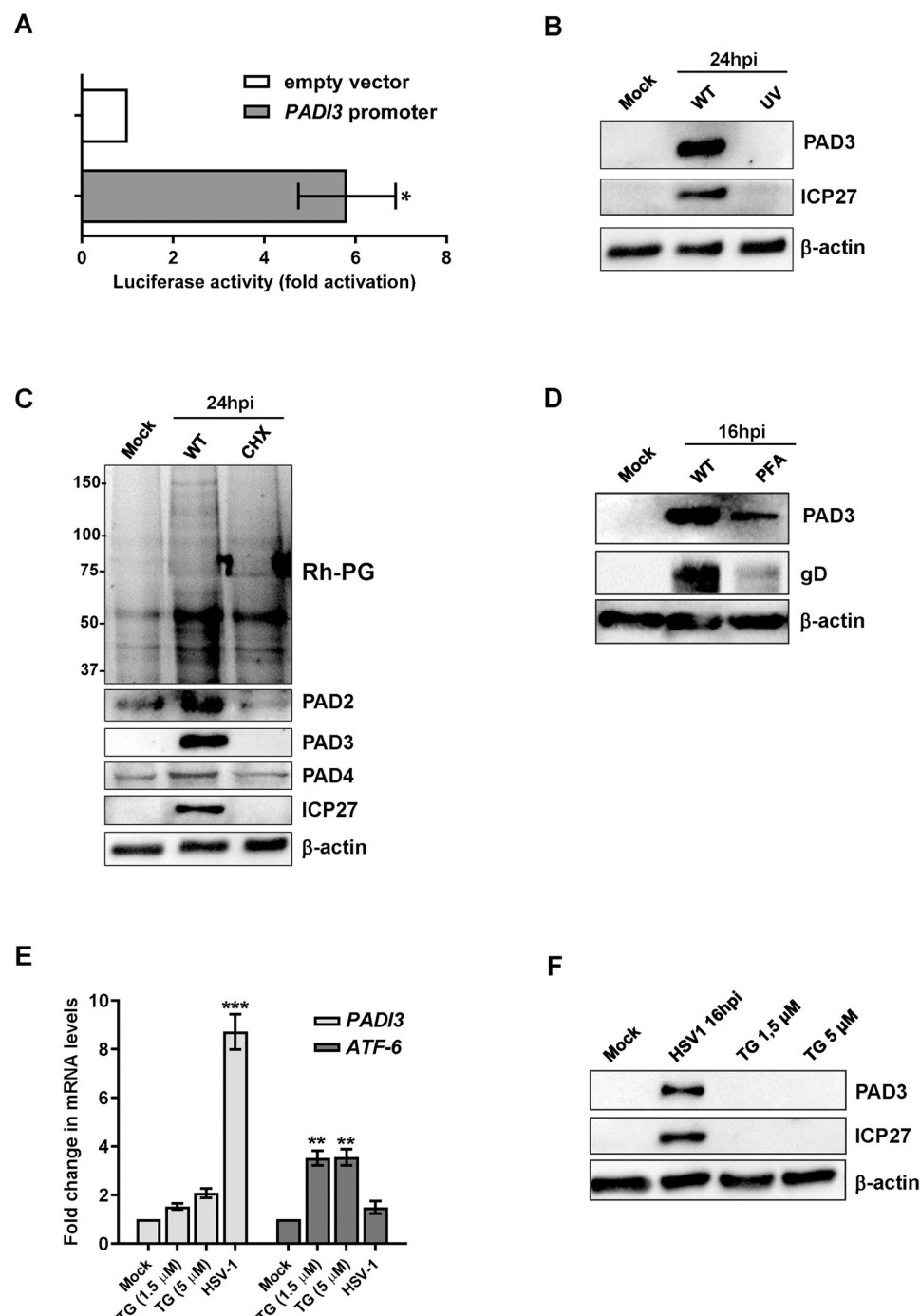

**Fig 3. PAD3 induction by HSV-1 early mechanism. (A)** HFFs were transiently transfected with luciferase plasmid encoding the wild-type PADI3 promoter region (pGL4.20-PADI3) or the pGL4.20 empty vector. Twenty-four h later, the cells were mock-infected or infected with HSV-1 at an MOI of 1. At 24 hpi, firefly and *Renilla* luciferase activities were measured. Luciferase activity in whole-cell lysates was normalized to *Renilla* luciferase activity and plotted as folds induction relative to infected HFFs carrying the pGL4.20 empty vector (set at 1). Results are shown as means of fold change ± SEM (error bars) of three independent experiments. **(B)** Western blot analysis of protein lysates from uninfected (mock) or infected HFFs with HSV-1 wild-type (WT) or UV-inactivated HSV1 (UV), at 24 hpi (MOI 1). Analysis was performed using antibodies against PAD3, ICP27, or β-actin. One representative blot of three independent experiments is shown. **(C)** Western blot and Rh-PG analysis of protein lysates from uninfected (mock) or HSV1-infected HFFs (MOI 1), treated with 150 μg/ml CHX or left untreated. Analysis was performed using antibodies against PAD2, PAD3, PAD4, ICP27, or β-actin to assess for equal loading. One representative blot of three

independent experiments is shown. **(D)** Protein lysates from uninfected (mock) or infected HFFs (16 hpi) at an MOI of 1 PFU/cell treated with (PFA 250 μM) or vehicle were analyzed by immunoblotting for PAD3, viral protein gD, or β-actin. **(E)** *PAD3* and *ATF-6* mRNA expression levels by RT-qPCR of HSV-1-infected (16 hpi, MOI 1) or mock-infected HFFs treated or not with the indicated amounts of thapsigargin (TG). The results were normalized to the housekeeping gene glyceraldehyde-3-phosphate dehydrogenase (GAPDH) and expressed as mean fold change ± SEM over mock-infected cells. $*P < 0.05$, $**P < 0.01$, $***P < 0.001$; one-way ANOVA followed by Bonferroni's post test. **(F)** Western blot analysis of protein lysates from uninfected (mock), HSV-1-infected (16 hpi, MOI 1) or mock-infected HFFs treated with TG for 16 h. Analysis was performed using antibodies against PAD3, ICP27, or β-actin. Representative blots of three independent experiments are shown.

## PAD3 targeting impairs HSV-1 replication

Having observed that PAD2, 3, and 4 protein levels increased during HSV-1 infection, we asked whether inhibiting the enzymatic activity of these proteins would affect viral replication. To this end, HFFs were treated with increasing concentrations of the specific inhibitors of the three isoforms (AFM30a for PAD2, HF4 for PAD3, and GSK199 for PAD4) or with equal volumes of vehicle control (DMSO) 1 h before HSV-1 infection (MOI 1) and for the entire duration of the infection. After 24 h of continuous exposure to the PAD inhibitors, we measured total virus production by plaque assay. While AFM30a and GSK199 treatment had no or minimal effect on HSV-1 replication rate at high doses (S3A and S3B Fig, respectively), exposure to HF4 strongly inhibited viral production in a dose-dependent manner, achieving a reduction of almost 3 logs at 5 μM (Fig 4A). To rule out the possibility that this antiviral activity was related to any off-target effects, we confirmed these results with another PAD3-specific inhibitor, namely CAY10727 [35], which showed a very similar efficacy in reducing viral production (Fig 4B). MTT assay excluded cytotoxic effect of drugs under the same treatment conditions (S3C, S3D and S3E Fig). In line with these results, at 24 hpi, infected carrier-treated cells showed a dramatic cytopathic effect, while HF4 and CAY10727 treatments restored a phenotype more similar to that of uninfected cells (Fig 4C). Moreover, immunoblot analysis of total protein extracts showed downregulation of viral late gD protein expression levels at 24 hpi in treated cells, particularly evident in the presence of HF4 and consistent with the inhibition observed in the plaque assay (Fig 4D). To corroborate these findings, we tested the PAD3 specific inhibitors HF4 and CAY10727 in the other cell types included in this study. As shown in Fig 4E and 4F, we observed a reduction in the count of viral particles in all cell lines in a dose-dependent manner, with the concentrations used being non-toxic for the cells (S3C and S3D Fig).

Finally, to confirm these results we knocked down PAD3 expression in HFF cells with a specific cocktail of PAD3-siRNAs and with siRNA control (siCtrl). At 24 h post-electroporation, cells were infected with HSV-1 (MOI 1). Protein lysates were harvested from siRNA-transfected cells at 24 hpi, and the efficiency of cellular gene expression reduction was evaluated at the protein level by Western blot analysis. We noted that PAD3 depletion was accompanied by a significant decrease in gD expression compared to control (Fig 4G). Moreover, the plaque assay on infected and PAD3-silenced HFFs showed a significant reduction of the virus yield by more than 2 logs compared to siCtrl HFFs (Fig 4H), which was confirmed by the complete recovery from the cytopathic effect upon PAD3 silencing (Fig 4I).

Overall, these data suggest that PAD3 activity plays a key role in supporting HSV-1 replication, and that PAD3 might be a promising target for anti-HSV-1 therapy.

## Citrullinome analysis reveals IFIT proteins as restriction factors for HSV-1 replication

Next, to identify which cellular and/or viral proteins were citrullinated during infection, we used a citrulline specific probe (biotin-PG) and enriched the HSV-1-associated citrullinome in

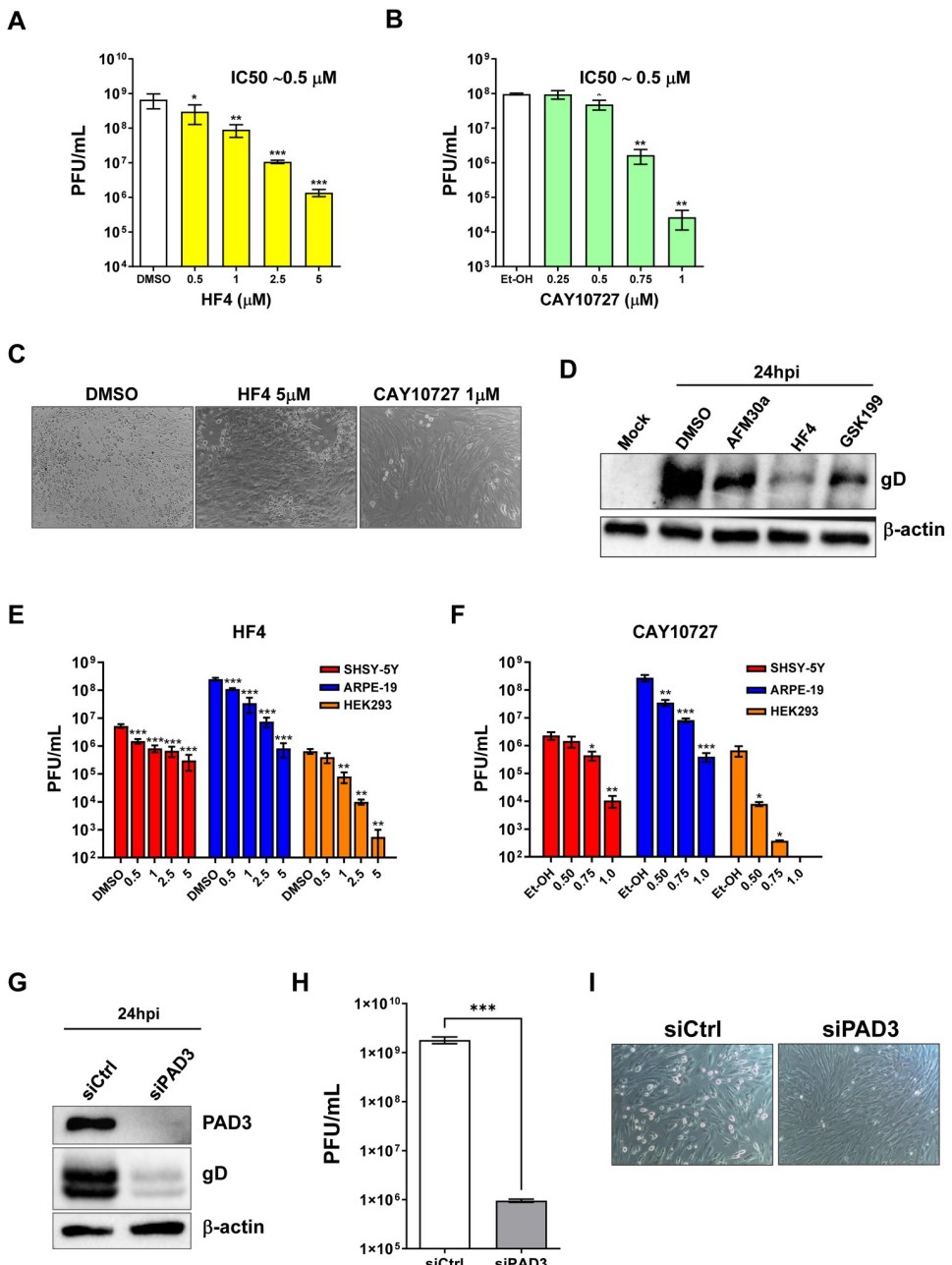

**Fig 4. PAD3 targeting impairs HSV-1 replication in human cells.** HFFs were infected with HSV-1 (MOI 1) and then treated with increasing concentrations of HF4 (**A**) or CAY10727 (**B**), two PAD3-specific inhibitors, which was given 1 h prior to virus adsorption and kept throughout the whole experiment. At 24 hpi, viral plaques were microscopically counted, and the number of plaques was plotted as a function of inhibitor concentration. Values are expressed as means ± SEM (error bars) of three independent experiments. Values are expressed as mean ± SEM of three independent experiments, *$P < 0.05$, **$P < 0.01$, ***$P < 0.001$; one-way ANOVA followed by Bonferroni's post test. (**C**) Representative images of infected HFFs (24 hpi) at an MOI of 1 PFU/cell and treated with HF4 (5 μM), CAY10727 (1 μM), or vehicle (DMSO). (**D**) Protein lysates from uninfected (mock) or infected HFFs (24 hpi) at an MOI of 1 PFU/cell treated with AFM30a (20 μM), HF4 (5 μM), GSK199 (20 μM) or vehicle (DMSO) were analyzed by immunoblotting to assess for viral expression with an anti-gD antibody; β-actin cellular expression was used for protein loading control. SH-SY5Y, ARPE-19, and HEK293 cells were infected with HSV-1 (MOI 1 PFU/cell) and then treated with increasing concentrations of HF4 (**E**) or CAY10727 (**F**), which were given 1 h prior to virus adsorption and kept throughout the whole experiment. At 24 hpi, viral plaques were microscopically counted, and the number of plaques was plotted as a function of inhibitor concentration. Values are expressed as means ± SEM (error bars) of four independent experiments, *$P < 0.05$, **$P < 0.01$, ***$P < 0.001$; one-way ANOVA followed by Bonferroni's post test.

(**G**) HFFs were silenced for PAD3 using specific siRNAs (siPAD3), as negative control cells were also similarly transfected with scrambled siRNA (siCTRL). At 24 h post-treatment (hpt), cells were infected with HSV-1 at an MOI of 1 PFU/cell. The efficiency of PAD3 protein depletion at 24 hpi was assessed by immunoblotting using antibodies against PAD3 or β-actin for equal loading. An anti-gD antibody was used to verify HSV-1 infection. Representative blots of three independent experiments are shown. (**H**) PAD3-silenced cells were infected with HSV-1 at an MOI of 1 PFU/cell. Viral supernatants were collected at 24 hpi and analyzed by standard plaque assay. Values are expressed as means ± SEM. Values are expressed as mean ± SEM of three independent experiments, ***$P < 0.001$; one-way ANOVA followed by Bonferroni's post test. (**I**) Representative images of infected HFFs (24 hpi) at an MOI of 1 PFU/cell and transfected with the same siCTRL and siPAD3 described in the legend to Fig 3D.

HFFs harvested at 16 and 24 hpi through streptavidin-agarose beads. As shown in Fig 5A, and consistent with our earlier findings, we observed a massive increase in overall protein citrullination in HSV-1-infected cells compared to uninfected control cells. In detail, database searches using the SEQUEST algorithm allowed us to identify 297 (126 significant) and 287 (125 significant) citrullinated cellular proteins at 16 and 24 hpi, respectively (S1 Data). In addition, a total of 39 and 35 (28 and 25 significant, respectively) citrullinated viral proteins were also detected at the indicated time points (Fig 5B and S2 Data). Of note, at 16 hpi most of the citrullinated viral proteins were immediate early (IE) genes and enzymes involved in the synthesis of viral DNA, whereas at 24 hpi mainly structural capsid and tegument proteins or envelope glycoproteins were detected. Moreover, through PANTHER software, we were able to identify a wide range of citrullinated host proteins falling into various functional classes, including metabolite interconversion enzymes, RNA and DNA metabolism proteins, chaperones, cytoskeletal proteins, protein-binding activity modulators, protein modifying enzymes, membrane traffic proteins, transporters, and defense/immunity proteins (Fig 5C).

Previously, we demonstrated that during HCMV infection several members of the interferon (IFN)-induced protein with tetratricopeptide repeat (*IFIT*) family were heavily citrullinated, and that IFIT1 lost its restriction factor activity when citrullinated *in vitro* [30]. Consistently, here we found that IFIT1 and IFIT2 were also significantly citrullinated in the HSV-1-associated citrullinome at both 16 and 24 hpi (Fig 5A). Unlike IFIT1 and IFIT2, IFIT3 showed only a slight increase in citrullination during infection, which was not statistically significant ($p = 0.06$). Therefore, we decided to focus on the other two isoforms for further investigation.

To validate these findings, total proteins from mock or HSV-1-infected HFFs at 16 and 24 hpi (MOI 1) were immunoprecipitated with an anti-citrulline antibody and subjected to immunoblotting using antibodies against IFIT1, IFIT2, or the viral protein ICP27. As shown in Fig 5D, IFIT1, IFIT2 and ICP27 proteins were upregulated and, consistent with previous results, citrullinated following infection with HSV-1, especially at 16 hpi. Considering that IFIT family members are significantly upregulated by IFNs, we sought to investigate whether the observed increase in citrullination was a direct consequence of the infection or resulted from the IFN-mediated upregulation in response to the infection. To test whether interferon alone could induce deimination of IFIT1 and IFIT2, we treated HFFs with IFN-β (500 U/mL) for 16 h and 24 h and immunoprecipitated the total proteins with the anti-citrulline antibody. Successively, we performed immunoblotting using antibodies against IFIT1 and IFIT2. As shown in Fig 5E, we found that both proteins were strongly upregulated by IFN treatment, but they did not bind to the anti-citrulline antibody, suggesting that interferon itself is not sufficient to cause deimination of IFIT proteins.

Next, to gain further insight into the role of these genes during HSV-1 infection, we measured viral production in HSV-1-infected HFFs after siRNA-mediated depletion of IFIT1 and IFIT2. Following transfection with specific siRNAs (siIFIT1 and siIFIT2, respectively) and

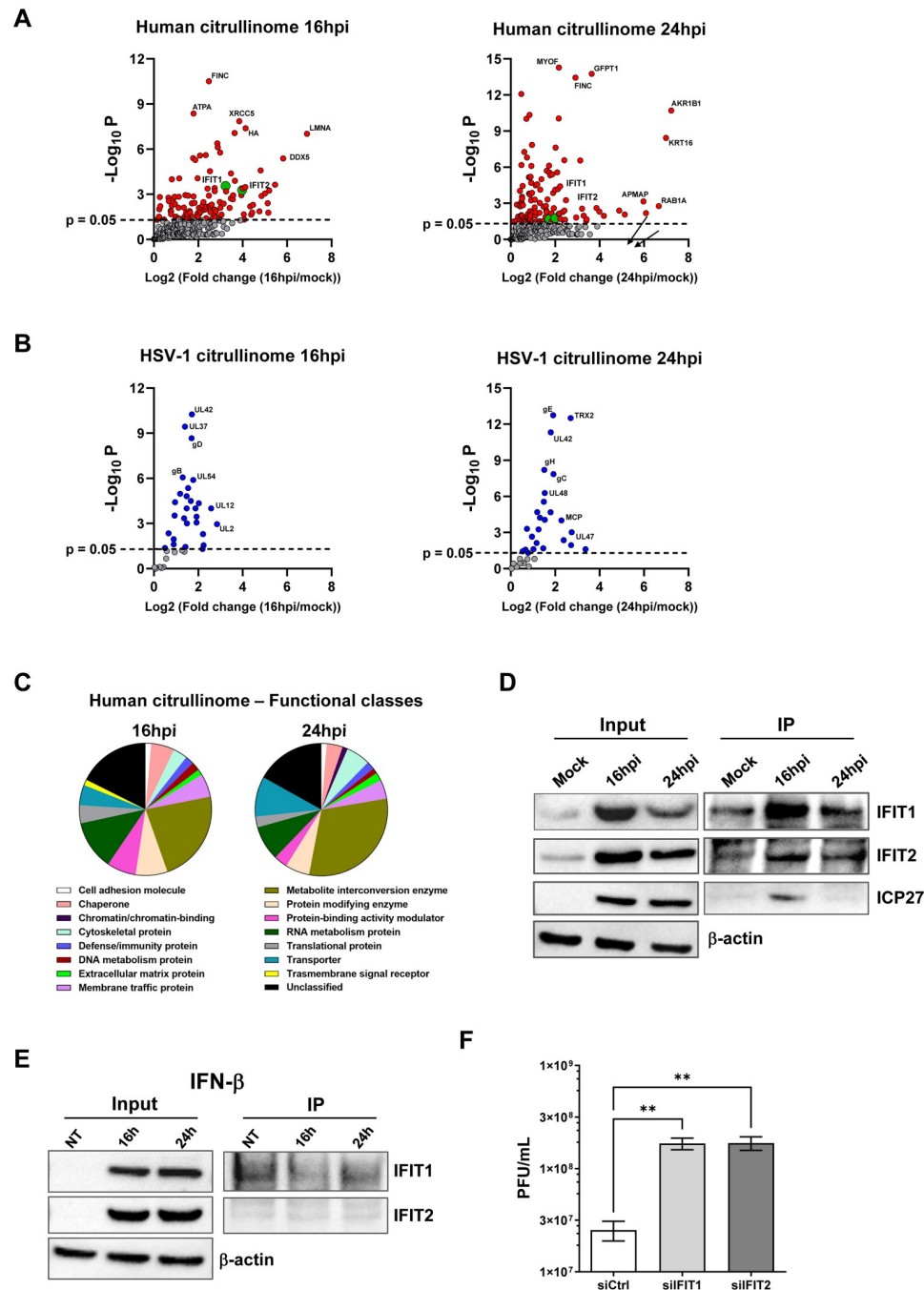

**Fig 5. Citrullinome analysis reveals IFIT proteins as restriction factors for HSV-1 replication. (A-B)** Volcano plot depicting the host (**A**, red dots) and viral (**B**, blue dots) citrullinated proteins of infected cells *vs* mock-infected cells at 16 hpi (left panels) and 24 hpi (right panels). Cell lysates from uninfected (mock) or HSV-1-infected HFFs (MOI 1) were exposed to a biotin-PG to isolate citrullinated proteins on streptavidin agarose. Bound proteins were then subjected to on-bead tryptic digestion and analyzed by LC-MS/MS—in the graph, every identified citrullinated protein corresponds to a dot. The x-axis represents the ratio of citrullination between mock and infected cells at the indicated time points, while the y-axis indicates the statistical significance. Both variables were plotted on a logarithmic scale (*n* = 3). IFIT1 and IFIT2 proteins are reported as green dots, while grey dots represent proteins that do not reach statistical significance. **(C)** The pie charts show the classification of citrullinated cellular proteins at 16 hpi (left) and 24 hpi (right) based on protein classes. Only classes with a representation exceeding 1% are reported. **(D)** Immunoprecipitation (IP) of total cell extracts (Input) from mock or infected HFFs at 24 hpi using an anti-peptidylcitrulline antibody. The IP complexes were analyzed by Western blotting using antibodies against IFIT1 and

IFIT2. An anti-ICP27 antibody was used to assess HSV-1 infection, while β-actin cellular expression was used for protein loading control. The blot shown is representative of three independent experiments. **(E)** Immunoprecipitation (IP) of total cell extracts (input) from untreated (NT) or IFN-β-treated (500U/ml) HFFs using an anti-peptidylcitrulline antibody. The IP complexes were analyzed by Western blot using antibodies against IFIT1 and IFIT2. Equal loading was assessed by β-actin immunoblotting. The blot shown is representative of three independent experiments. **(F)** HFFs were silenced for IFIT1 and IFIT2 using specific siRNAs (siIFIT1 and siIFIT2, respectively). As negative control cells were also similarly transfected with scrambled siRNA (siCtrl. At 24 hpt, cells were infected with HSV-1 at an MOI of 1 PFU/cell. Viral supernatants were collected at 24 hpi and analyzed by standard plaque assay. Values are expressed as means ± SEM of three independent experiments. Representative blots of three independent experiments are shown.

subsequent infection with HSV-1 for 24 h, we achieved a nearly complete silencing of both proteins, without affecting the ability of the virus to replicate, as demonstrated by the expression of the ICP27 viral marker (S4A Fig). Rather, silencing either IFIT1 or IFIT2 resulted in significantly higher levels of virus production compared to siCtrl-transfected cells (Fig 5F), suggesting that IFIT1 and IFT2 are potential restriction factors for HSV-1, whose activity can be overcome by virus-induced citrullination.

## Discussion

Viruses, unlike other infectious agents, require the molecular machinery of the host to complete their replication cycle. As a result, these pathogens have evolved multiple mechanisms to promote the translation of their transcripts, evade the immune system, and induce host gene expression alterations during infection. These mechanisms are indispensable for the viral life cycle, but they are particularly sophisticated in viruses, such as herpesviruses, that establish latency and persist throughout the host's lifespan. Recent research has unveiled the potential involvement of host-cellular environment manipulation mechanisms in the pathogenesis of autoimmune diseases, as well as tumors and neurodegenerative disorders. A promising strategy to target these evasion mechanisms is to employ host-targeting antiviral agents (HTAs) that interfere with the host factors required for viral replication [36].

Citrullination, a post-translational modification mediated by PAD enzymes, is emerging as a key mechanism that viruses exploit during their replication in host cells. This is supported by an association between viral infection and PAD-mediated upregulation of citrullination, which we and others have reported in various viral and cellular models [25–27,30]. In particular, we have shown that one such virus, HCMV, induces PAD-mediated citrullination of several cellular proteins endowed with antiviral activity, including the IFN-stimulated genes (ISGs) IFIT1 and Mx1, and that blocking this process with the PAD inhibitor Cl-amidine can inhibit viral replication [30]. Here, we extend those findings to a different member of the *Herpesviridae* family, HSV-1, which we demonstrate to be capable of promoting PAD-mediated citrullination *in vitro*. Specifically, we show that HSV-1 infection of human fibroblasts upregulates total protein citrullination, and that this process is required for optimal viral replication.

Despite various viruses being capable of upregulating PAD enzyme expression, which is crucial for optimal viral replication, our findings suggest that each virus displays a preference for specific PAD isoforms. This challenges the assumption that PAD upregulation is a general response to viral infection. For instance, PAD4 is the main isoform regulated by human rhinovirus (HRV) and HCoV [26,27], while PAD2 and PAD4 cooperate in supporting HCMV replication [30]. In the present work, we show that during HSV-1 infection three different PAD isoforms are upregulated: PAD2, 3, and 4. Although PAD2 and PAD4 are also induced upon HSV-1 replication, PAD3 seems to play a major role, as shown by the fact that its promoter is robustly induced by HSV-1 infection and that its targeted inactivation

through specific inhibitors or silencing by siRNAs has a detrimental effect on HSV-1 replication. Noteworthy, during HSV-1 infection PAD3 consistently displayed similar kinetics to that of the viral protein ICP27, regardless of the cell model used, suggesting that the variations in permissiveness among cells, and consequently, the different kinetics of viral replication, rather than the different basal expression of PAD3, influence the kinetics of PAD expression. Hence, developing specific inhibitors for this particular isoform could improve the efficacy and mitigate the toxicity of HSV-1 antiviral treatments. This underscores the overarching need to identify which isoforms are specifically involved in different viral infections to design more effective HTAs [37].

In this study, we have also identified the cellular and viral proteins deiminated during HSV-1 infection, thus determining the citrullinome of the infected cells. We show that several members of the interferon (IFN)-induced protein with tetratricopeptide repeat (*IFIT*) family, such as IFIT1 and IFIT2, are robustly citrullinated, similar to what we observed in HCMV-infected cells. This is of particular interest because IFIT family members, primarily known for their antiviral activity against RNA viruses [38,39], have recently been linked to the innate immune response against DNA viruses, including two other members of the herpesvirus family, namely KSHV and HCMV [30,40]. More specifically, the antiviral function of IFIT3—which in our analysis was not significantly citrullinated—against HSV-1 is known, as is the ability of the virus to evade this function via the UL41 protein [41]. However, the roles of the other two members, IFIT1 and IFIT2, in HSV-1 infection have not been previously reported. Our research shows that silencing of either IFIT1 or IFIT2 results in increased virus production, while ectopic expression of IFIT1 inhibits viral growth, which agrees with previous findings in KSHV and HCMV-infected cells [30,40]. In terms of viral proteins, our study reveals that citrullination follows the natural replication cycle of the virus, with IE, E, and L proteins undergoing citrullination in a sequential order. Importantly, both viral structural and catalytic proteins show citrullination, underscoring the need to understand how this modification impacts their functions and whether it is involved in the assembly of new virions. In this regard, the substantial presence of citrullinated cellular and viral proteins observed in our analysis indicates that infection-induced citrullination is critically involved in multiple stages of HSV-1 replication throughout its life cycle. Therefore, PAD family members represent promising candidates for antiviral drug targeting, as their inhibitors can effectively block the replication of HSV-1 through various mechanisms.

It is however worth pointing out that although some substrates of PADs are IFN-inducible proteins, citrullination *per se* is not an IFN-dependent phenomenon. We show that, in mock-infected cells exposed to interferon, IFIT1 and IFIT2 proteins are upregulated but not over-citrullinated, and that the expression levels of PAD enzymes are unchanged. These findings, along with the fact that PADs are only upregulated in cells infected with functional HSV-1 but not in cells infected with UV-treated HSV-1, imply that viral IE protein production regulates PAD protein expression, as we observed in HCMV-infected cells [30].

Overall, our results clearly demonstrate that PAD induction and subsequent citrullination profile alteration are virus-specific phenomena. However, it is highly likely that the over-citrullinated host proteins in the course of infection are closely dependent on the infected tissue, just as the citrullinome is strictly linked to the gene expression profile of host cells prior to infection [42]. Consequently, it is highly probable that the effects of citrullination induced by an infection will vary depending on the specific tissue and organ affected, rather than simply being influenced by the type of virus involved.

Altogether, our findings suggest the intriguing scenario that conditions where citrullination contributes to a pathological state may be triggered by a viral infection that initially disrupts PAD activity.

## Materials and methods

### Cell lines and viruses

Human foreskin fibroblasts (HFFs; ATCC, SCRC-1041), African green monkey kidney Vero cells (Sigma-Aldrich, 84113001), the human neuroblastoma cell line SH-SY5Y (ATCC, CRL-2266), human embryonic kidney cells HEK293 (ATCC, CRL-1573) and retinal pigment epithelial cell (ARPE-19; ATCC, CRL-2302) were grown in Dulbecco's Modified Eagle Medium (DMEM; Sigma-Aldrich) supplemented with 1% (v/v) penicillin/streptomycin solution (Euroclone) and heat-inactivated 10% (v/v) fetal bovine serum (FBS) (Sigma-Aldrich). The medium of SH-SY5Y cells was also supplemented with non-essential amino acids (NEAA, Sigma-Aldrich). The clinical isolate of HSV-1 was grown in Vero cells and titrated by standard plaque assay as described previously [43].

### Reagents and treatments

The PAD inhibitors Cl-amidine (Cl-A), BB-Cl-amidine (BB-Cl), GSK199, CAY10727, and AFM30a—also known as CAY10723—were obtained from Cayman Chemical (Ann Arbor). CHX and foscarnet (phosphonoformic acid, PFA) were from Sigma-Aldrich. All these compounds were reconstituted in dimethyl sulfoxide (DMSO) or ethanol accordingly to their solubility. HF4 and Thapsigargin were kindly provided by Dr. P. Thompson and Dr. M. Corazzari, respectively. Immediately before use, the inhibitors were diluted in the cell medium to their final concentrations. For the PADs inhibitors experiments, cells were pre-treated with the inhibitors for 1 h and then infected (MOI 1) by adding the virus without changing the medium. Following virus adsorption (2 h), the viral inoculum was removed, and a new medium with fresh inhibitor was added. After that, no more medium or inhibitor was added until the samples were collected.

### Cell viability assay

Cells were seeded at a density of $3*10^4$/well in a 96-well plate. After 24 h, the cells were treated with the indicated concentration of the different inhibitors or mock-treated using the vehicle alone (DMSO or ethanol). After 48 h, cell viability was determined using the 3-(4,5-dimethylthiazol-2-yl)-2,5-diphenyltetrazolium bromide (MTT, Sigma) method as described previously [30].

### *In vitro* antiviral assay

Cells were cultured in a 24-well plate for 1 day and then treated as described in 3.2. Briefly, they were incubated with the aforementioned PAD inhibitors at the indicated concentration for 1 h and subsequently infected with HSV-1 at a MOI of 1. Following virus adsorption (2 h at 37°C), the viral inoculum was removed, and the cell cultures were maintained in a medium containing the indicated treatment for 24 h. DMSO or ethanol were used as a negative control. Next, the cells and supernatants were harvested, pooled, and then lysed by two freeze-thaw cycles. The extent of virus replication was then assessed by titrating the infectivity of the sample by standard plaque assay on Vero cells.

### Plaque assay

Vero cells were seeded at a density of $3*10^4$/well in a 96-well plate and inoculated 24 h later with 10-fold serial dilutions of the HSV-1 production. After 48 h, cells were fixed and stained with crystal violet solution, and plaques were counted on each well to determine the virus titer,

which was determined by counting the number of immunostained foci on each well using the following formula: virus titer (PFU/ml) = number of plaques * 0.1 ml/dilution fold.

## DNA and RNA isolation and quantitative nucleic acid analysis

Total DNA and RNA were extracted using the NucleoSpin RNA kit (Macherey-Nagel, Düren, Germany), and 1 μg of RNA was retrotranscribed using the Revert-Aid H-Minus FirstStrand cDNA Synthesis Kit (Thermo Fisher Scientific, Waltham, USA), according to the manufacturer's instructions. Comparison of mRNA expression between samples (*i.e.*, infected *vs*. untreated) was performed by SYBR green-based RT-qPCR using a Biorad CFX96 apparatus, using the primers reported in S1 Table. To determine the number of viral DNA genomes, viral DNA levels were measured by quantitative PCR on a Biorad CFX96 apparatus (Stratagene, San Diego, USA). To create a standard curve for each analysis, genomic DNA mixed with a gE2-encoding plasmid (Addgene pAcUW51-gE2) was serially diluted from 109 to 1 copy and analyzed in parallel. The amount of human GAPDH gene amplified per reaction mixture was used to normalize the HSV-1 DNA copy numbers.

## Western blot analysis

Cells were treated with the indicated compound or equal volumes of DMSO solvent 1 h before infection and throughout the entire duration of the infection as previously described in point 2.2. Cells were infected with HSV-1 at an MOI of 1 and harvested at the indicated time points. Cell lysates were prepared with RIPA buffer, quantified by Bradford method, and subjected to Western blot analysis. The primary antibodies were as follows: anti-PAD1 (Abcam, ab181791); anti-PAD2 (Cosmo Bio, SML-ROI002-EX); anti-PAD3 (Abcam, ab50246); anti-PAD4 (Abcam, ab128086); anti-PAD6 (Abcam ab16480); anti-actin (Sigma-Aldrich, A2066), anti-gD (Virusys, HA025-1), anti-ICP27 (Virusys, P1113), anti-IFIT1 (Invitrogen PA5-31254), anti-IFIT2 (Proteintech, 12604-1-AP), anti-IFIT3 (Proteintech 15201-1-AP).

## Detection of citrullination with rhodamine-phenylglyoxal (Rh-PG)

Whole-cell protein extracts were prepared as described in the previous paragraph. Equal amounts of protein were diluted with 80% trichloroacetic acid and incubated with Rh-PG (final concentration 0.1 mM) for 30 min [44]. The reaction was quenched with 100 mM L-citrulline, then centrifuged at 21,100xg for 10 min, washed with ice-cold acetone, and resuspended in 2X SDS loading dye for gel electrophoresis. Gels were imaged (excitation = 532 nm, emission = 580 nm) using a Biorad Chemidoc Imaging System, stained with brilliant blue G-colloidal solution (Sigma-Aldrich).

## PAD3 vector construction

The 5' flanking region of *PADI3* gene was amplified by PCR using the Q5 High-Fidelity DNA polymerase (New England Biolabs, Ipswich, USA), the human genomic DNA from HFFs as a template, and the PAD3 primers containing *Xho* I and *Hind* III restriction enzymes sites (see primers sequences listed in S1 Table). The resulting amplification products were digested with *Xho* I and *Hind* III (Thermo Fisher Scientific, Waltham, USA) and cloned into the pGL4.20 [luc2/Puro] vector (Promega, Madison, USA), which encodes the luciferase reporter gene *luc2* (*Photinus pyralis*) with no other regulatory elements. The resulting pGL4.20-PADI3prom construct was purified using the PureYield Plasmid Miniprep System (Promega, Madison, USA) and verified by restriction mapping and complete sequencing. The resulting chromatograms were analyzed using Chromas software 2.6.6 (Technolysium Ltd.).

## Dual-luciferase reporter assay

HFFs were seeded into 12-well plates and, after 24 h of incubation, transiently transfected with 1 μg of pGL4.20 or pGL4.20-PADI3 plasmids together with 0.1 μg of Renilla reporter plasmid (pRL-SV40, to correct for transfection efficiency) using Lipofectamine 2000 Transfection Reagent kit (Thermo), according to the manufacturer's instruction. After 24h, cells were infected with HSV-1 (MOI of 1 PFU/ml), and 24 hpi the Dual-luciferase Reporter Assay System (Promega, USA) was used to detect Firefly and *Renilla* luciferase activities and recorded using GloMax 96 Microplate Luminometer (Promega, USA). Firefly luciferase activity from the luciferase reporter vector was normalized to *Renilla* luciferase activity from the pRL-SV40 vector and plotted as folds of induction relative to infected HFFs expressing the pGL4.20 empty vector (set at 1).

## Pull-down experiments

Uninfected or HSV-1-infected cells (MOI of 1 PFU/ml) were washed with 1X PBS and lysed in radioimmunoprecipitation assay (RIPA) buffer (50 mM Tris pH 7.4; 150 mM NaCl; 1 mM EDTA; 1% nonidet P-40; 0.1% SDS; 0.5% deoxycholate; protease inhibitors). Proteins (200 μg) were then incubated with 2 μg of anti-citrulline monoclonal antibody (Cayman Chemical, 30773) or with an isotype antibody as negative control (62–6820; Thermo Fischer Scientific, Waltham, USA) for 1 h at room temperature with rotation followed by overnight incubation at 4°C with protein G-sepharose (Sigma-Aldrich, Milan, Italy). Immune complexes were collected by centrifugation and washed with RIPA buffer. The sepharose beads were pelleted and washed three times with RIPA buffer, resuspended in reducing sample buffer (50 mM Tris pH 6.8; 10% glycerol; 2% SDS; 1% 2- mercaptoethanol), boiled for 5 min and resolved on an SDS-PAGE gel to assess protein binding by immunoblotting.

## Citrullinome analysis by mass spectrometry: Sample preparation

Sample preparation in technical triplicates followed the procedure outlined in [45]. Equal amounts of cell lysates from each experimental group (300 μg) were diluted in buffer (100 mM HEPES pH 7.6) to a final concentration of 1 μg/μL and incubated with 20% trichloroacetic acid (TCA) and 0.5 mM biotin-PG [46] for 30 min at 37 °C. Labeled proteomes were precipitated on ice for 30 min. Samples were pelleted through tabletop centrifugation (21,100xg, 15 min) at 4 °C. The supernatants were discarded, and the pellets were washed with cold acetone (300 μL). After drying for 5 min, the pellets were resuspended in 1.2% SDS in PBS by bath sonication and heating. Samples were then transferred to 15 mL screw cap tubes and diluted in 1X PBS to a 0.2% SDS final concentration. Samples were incubated with streptavidin agarose slurry (Sigma Aldrich, 170 μL) overnight at 4 °C and for an additional 3 h at 25 °C. After discarding the flow through, the streptavidin beads were washed with 0.2% SDS in PBS (5 mL) for 10 min at 25 °C. The beads were then washed three times with 1X PBS (5 mL) and three times with water (5mL) to remove any unbound proteins. Beads were then transferred to a screw cap microcentrifuge tube and heated in 1X PBS with 500 μL 6 M urea and 10 mM DTT (65 °C, 20 min). Proteins bound to the beads were then alkylated with iodoacetamide (20 mM, 37 °C for 30 min). The beads were successively pelleted by centrifugation (1,400 x *g* for 3 min) and the supernatant was removed. The pellet was resuspended in a premixed solution of 2 M urea, 1 mM CaCl$_2$ and 2 μg Trypsin Gold (Promega, Madison, USA) in PBS. These were shaken overnight at 37 °C. The supernatant was collected, and the beads were washed twice with water (50 μL), each time collecting the supernatant. The fractions were combined, acidified with formic acid (5% final concentration) and stored at -20 °C until use.

## Mass spectrometry

Liquid chromatography-mass spectrometry/mass spectrometry (LC-MS/MS) analysis was performed with an LTQ-Orbitrap Discovery mass spectrometer (Thermo Fisher Scientific, Waltham, MA, USA) coupled to an Easy-nLC HPLC (Thermo Fisher Scientific, Waltham, MA, USA). Samples were pressure loaded onto a 250-μm fused-silica capillary hand packed with 4-cm Aqua C18 reverse phase resin (Phenomenex). Samples were separated on a hand packed 100-μm fused-silica capillary column with a 5-μm tip packed with 10 cm Aqua C18 reverse phase resin (Phenomenex). Peptides were eluted using a 10-h gradient of 0–100% Buffer B in Buffer A (Buffer A: 95% water, 5% acetonitrile, 0.1% formic acid; Buffer B: 20% water, 80% acetonitrile, 0.1% formic acid). The flow rate through the column was set to ~400 nL/min, and the spray voltage was set to 2.5 kV. One full MS scan (FTMS) was followed by 7 data-dependent MS2 scans (ITMS) of the $n^{th}$ most abundant ions. The tandem MS data were searched by the SEQUEST algorithm using a concatenated target/decoy variant of the human and viral UniProt database. A static modification of +57.02146 on cysteine was specified to account for alkylation by iodoacetamide. SEQUEST output files were filtered using DTASelect 2.0.

## Database search

Raw data were processed and searched using Maxquant 1.6.14 and its integrated Andromeda search engine using the Swiss-Prot human (downloaded 04/09/2019) and Uniprot HCMV (downloaded 02/27/2021). Search parameters were as follows: tryptic digestion with up to 2 missed cleavages; peptide N-terminal acetylation, methionine oxidation, N-terminal glutamine to pyroglutamate conversion were specified as variable modifications. The monoisotopic mass increment of the triplex dimethyl labels, light, medium and heavy dimethyl label at 28.0313, 32.0564 and 36.0757 Da, respectively, were set as variable modification on the peptide N-termini and lysine residues. Carbamidomethylation of cysteines was set as static modification. Main search tolerance was 6 ppm, and the first search tolerance was 50 ppm. Both the protein and peptides identification false discovery rates (FDR) were < 1%. Protein grouping, dimethyl ratio calculations and downstream statistics were performed in Scaffold Q+S 4.8.9 (Proteome Software, Portland, OR).

## siRNA-mediated knockdown

HFFs were transiently transfected with a Neon Transfection System (Thermo Fischer Scientific) according to the manufacturer's instructions (1200 V, 30 ms pulse width, one impulse) with a pool of small interfering RNAs targeting PAD3 (Sigma-Aldrich, EHU012711), IFIT1 (Qiagen S102660777), IFIT2 (Invitrogen AM16708) or control siRNA (siCTRL, Qiagen 1027292) as negative control.

## Statistical analysis

All statistical tests were performed using GraphPad Prism version 9.5.1 for Windows (GraphPad Software, San Diego, USA). The data were presented as means ± standard error of mean (SEM). Statistical significance was determined by using unpaired t test (two-tailed), one-way or two-way analysis of variance (ANOVA) with Bonferroni's, or Dunnett's post-tests. Differences were considered statistically significant for $P < 0.05$ (*$P < 0.05$; **$P < 0.01$; ***$P < 0.001$). The half-maximal inhibitory concentration ($IC_{50}$) was calculated by Quest Graph IC50 Calculator (AAT Bioquest, Inc, https://www.aatbio.com/tools/ic50-calculator).

## Supporting information

**S1 Fig.** Uninfected HFFs, ARPE-19, and SHSY-5Y were treated with the indicated concentrations of Cl-A (**A**) and BB-Cl (**B**) for 24 h, and the number of viable cells was determined for each concentration using the MTT assay. Values are expressed as means ± SEM of three independent experiments. SH-SY5Y (**C**) and ARPE-19 (**D**) were infected with HSV-1 (MOI 1 PFU/cell) and then treated with increasing concentrations of Cl-A, which were given 1 h prior to virus adsorption and kept throughout the whole experiment. At 24 hpi, viral plaques were microscopically counted, and the number of plaques was plotted as a function of inhibitor concentration. Values are expressed as means ± SEM (error bars) of three independent experiments, $*P < 0.05$, $**P < 0.01$, $***P < 0.001$; one-way ANOVA followed by Bonferroni's post-test. (**E**) Protein lysates from uninfected (mock) or infected HFFs (24 hpi) at an MOI of 1 PFU/cell treated with Cl-A (100 μM), BB-Cl (2.5 μM) or vehicle (DMSO) were exposed to an Rh-PG citrulline-specific probe (left panel) and subjected to gel electrophoresis to detect citrullinated proteins. Equal loading was assessed by Coomassie blue staining (right panel). (**F**) Representative images of infected HFFs (24 hpi) at an MOI of 1 PFU/cell and treated with Cl-A (100 μM), BB-Cl (2.5 μM), or vehicle (DMSO).
(TIF)

**S2 Fig.** (**A**) mRNA expression levels of *PADI* isoforms by RT-qPCR of HSV-1-infected (8 and 16 hpi) *vs* uninfected (mock) HFFs were normalized to the housekeeping gene glyceraldehyde-3-phosphate dehydrogenase (GAPDH) and expressed as mean fold change ± SEM over mock-infected cells. $*P < 0.05$, $**P < 0.01$, $***P < 0.001$; one-way ANOVA followed by Bonferroni's post test. (**B**) Densitometric analysis of three independent experiments, values are expressed as fold change in PAD2 and PAD4 expression normalized to α-tubulin. (**C**) mRNA expression levels of *PADI3* isoforms by RT-qPCR of HSV-1-infected (6, 16 and 24 hpi) *vs* uninfected (mock) cells were normalized to GAPDH and expressed as mean fold change ± SEM over mock-infected cells. $*P < 0.05$, $**P < 0.01$, $***P < 0.001$; one-way ANOVA followed by Bonferroni's post test. (**D**) Western blot analysis of protein lysates from untreated (NT) or IFN-β treated (500 U/mL) HFFs using antibodies against PAD1 or PAD6. β-actin cellular expression was used for protein loading control. One representative gel of three independent experiments is shown. (**E**) Blu Coomassie staining of the same protein extracts used in the experiments shown in Fig 2D (protein lysates from uninfected (mock) or HSV1-infected HFFs, treated with 150 μg/ml CHX or left untreated).
(TIF)

**S3 Fig.** (**A-B**) HFFs were infected with HSV-1 (MOI 1 PFU/cell) and then treated with increasing concentrations of AFM30a (A), GSK199 (B) and CAY10727 (C), which were given 1 h prior to virus adsorption and kept throughout the whole experiment. At 24 hpi, viral plaques were microscopically counted, and the number of plaques was plotted as a function of inhibitor concentration. Values are expressed as means ± SEM (error bars) of three independent experiments, $*P < 0.05$, $**P < 0.01$, $***P < 0.001$; one-way ANOVA followed by Bonferroni's post test. (**C-E**) Uninfected HFF, SHSY-5Y, ARPE-19 or HEK293 cells were treated with the indicated concentrations of HF4 (C), CAY10727 (D), AFM30a, or GSK199 (E) for 24 h and the number of viable cells was determined for each concentration by MTT assay. Values are expressed as means ± SEM of three independent experiments.
(TIF)

**S4 Fig.** (**A**) The efficiency of IFIT1 or IFIT2 protein depletion at 24 hpi was assessed by immunoblotting using antibodies against IFIT1 or IFIT2, or against β-actin to check for equal loading. An anti-ICP27 antibody was also used to verify HSV-1 infection. Representative blots of

three independent experiments are shown.
(TIF)

**S1 Table. Oligonucleotide primer sequences for qPCR and PADI3 promoter cloning.**
(PDF)

**S1 Data. List of the host citrullinated proteins of infected *vs* mock-infected HFFs at 16 hpi and 24 hpi.**
(XLSX)

**S2 Data. List of the viral citrullinated proteins of infected *vs* mock-infected HFFs at 16 hpi and 24 hpi.**
(XLSX)

## Acknowledgments

We thank Marcello Arsura for critically reviewing the manuscript, and Dr. M. Corazzari and Dr. Romina Monzani (CAAD Center, University of Piemonte Orientale, Novara) for providing reagents and expertise for the experiments with thapsigargin.

## Author Contributions

**Conceptualization:** Selina Pasquero, Francesca Gugliesi, Paul R. Thompson, Santo Landolfo, Marco De Andrea.

**Data curation:** Selina Pasquero, Francesca Gugliesi, Camilla Albano, Stefano Raviola, Sen Sudeshna, Marco De Andrea.

**Formal analysis:** Camilla Albano, Greta Bajetto, Linda Trifirò, Bianca Brugo, Qiao Yang, Sen Sudeshna, Leonard Barasa, Hafeez Haniff.

**Funding acquisition:** Francesca Gugliesi, Matteo Biolatti, Valentina Dell'Oste, Marco De Andrea.

**Investigation:** Selina Pasquero, Francesca Gugliesi, Camilla Albano, Greta Bajetto, Gloria Griffante, Linda Trifirò, Bianca Brugo, Davide Lacarbonara, Qiao Yang, Sen Sudeshna, Leonard Barasa.

**Methodology:** Selina Pasquero, Francesca Gugliesi, Matteo Biolatti, Valentina Dell'Oste, Camilla Albano, Greta Bajetto, Bianca Brugo, Stefano Raviola, Sen Sudeshna.

**Project administration:** Marco De Andrea.

**Resources:** Paul R. Thompson.

**Software:** Selina Pasquero, Francesca Gugliesi, Greta Bajetto, Gloria Griffante, Linda Trifirò, Davide Lacarbonara, Qiao Yang, Sen Sudeshna, Leonard Barasa, Hafeez Haniff, Paul R. Thompson.

**Supervision:** Matteo Biolatti, Valentina Dell'Oste, Paul R. Thompson, Marco De Andrea.

**Validation:** Selina Pasquero, Francesca Gugliesi, Matteo Biolatti, Valentina Dell'Oste, Stefano Raviola, Sen Sudeshna, Hafeez Haniff.

**Visualization:** Selina Pasquero, Francesca Gugliesi, Gloria Griffante, Leonard Barasa, Hafeez Haniff.

**Writing – original draft:** Selina Pasquero, Francesca Gugliesi, Valentina Dell'Oste.

**Writing – review & editing:** Paul R. Thompson, Santo Landolfo, Marco De Andrea.

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
