## [Decision Letter · Decision Letter 0]

17 Aug 2023

Dear Dr. De Andrea

Thank you very much for submitting your manuscript "Citrullination profile analysis reveals peptidylarginine deaminase 3 as an HSV-1 target to dampen the activity of candidate antiviral restriction factors" for consideration at PLOS Pathogens. As with all papers reviewed by the journal, your manuscript was reviewed by members of the editorial board and by several independent reviewers. In light of the reviews (below this email), we would like to invite the resubmission of a significantly-revised version that takes into account the reviewers' comments.

Three experts in the field have all agreed that the work presented in this manuscript was exciting, novel and of interest to a wide spectrum of researchers. However, there were several additional experiments that reviewers thought were required to further support the conclusions stated. This includes performing the experiments in an additional cell type that expresses basal levels of PAD3 and using foscarnet to differentiate IE/E gene expression. Additionally, it is also crucial that you address every comment or point made by each of the three reviewers in your revised manuscript.

We cannot make any decision about publication until we have seen the revised manuscript and your response to the reviewers' comments. Your revised manuscript is also likely to be sent to reviewers for further evaluation.

Sincerely,

Donna M Neumann

Academic Editor

PLOS Pathogens

Alison McBride

Section Editor

PLOS Pathogens

Kasturi Haldar

Editor-in-Chief

PLOS Pathogens

orcid.org/0000-0001-5065-158X

Michael Malim

Editor-in-Chief

PLOS Pathogens

orcid.org/0000-0002-7699-2064

Your manuscript was reviewed by three experts in the field and they all agreed that yje work was exciting and of interest to a wide spectrum of researchers. However, there were several additional experiments that reviewers thought were required to further support the conclusions stated. This includes performing the experiments in an additional cell type that expresses basal levels of PAD3 and using foscarnet to differentiate IE/E gene expression. Additionally, it is also crucial that you address every comment or point made by each of the three reviewers.

Reviewer's Responses to Questions

**Part I - Summary**

Reviewer #1: The manuscript, “Citrullination profile analysis reveals peptidylargine deaminase 3 as an HSV-1 target to dampen the activity of candidate antiviral restriction factors” by Pasquero and others, describes how HSV-1 specifically up-regulates expression of the 3rd isoform of the peptidylarginine deaminases (PADs) during infection. This is different than HCMV, SARS-CoV2 and rhinovirus, which up regulate different PAD isoforms during infection. Using PAD3 specific inhibitors during infection, the authors show that PAD3 is required by the virus for robust viral replication and late gene expression, and for citrullinating important interferon (IFN)-induced genes, like IFIT1 and IFIT2, which inhibits their function and promotes virus replication. The authors use this information to support the development of host-targeting antivirals (HATs) and suggest that a PAD3 inhibitor would be specific and relevant as an anti-HSV-1 therapeutic.

In general, the claims made in this paper are original only if PAD3 is shown to be specifically regulated by HSV-1 infection, and this will be of broad interest to the wider community of researchers studying pathogens and host-pathogen interactions. It is very novel that HSV-1 seems to choose PAD3 specifically, while other viruses use different isoforms. However, these claims require a much more intensive investigation with a cell type that has detectable basal levels of PAD3, and other HSV-1 relevant cell types. Additionally, a different housekeeping gene other than GAPDH or a different approach to measuring cellular transcript levels needs to be employed for the qRTPCR experiments, to overcome the well documented virion host shut off activity of HSV-1. These comments and others are outlined below in more depth.

Reviewer #2: In an excellent submission, Pasquero et al. demonstrate that PAD inhibitors block HSV infection, and that HSV infection results in citrullination of virus and host proteins. IFIT1 and IFIT2 are both citrullinated during infection, and knocking them down increases HSV infection. The authors propose that HSV-triggered citrullination of host restriction factors is a novel way for HSV to evade the innate immune response. The paper is well-written and easy to follow. The data are of high quality, and the data interpretation is on point. The findings are compelling, novel, and move the field forward. A logical next step would be to determine whether deimination of IFIT1 and IFIT2 affects their ability to restrict HSV infection. If so, then this would round out a mechanism for the role of citrullination in HSV infection. I think this is best left for the future given there are a lot of new, significant results here. As a result of the current work, there is a solid hypothesis to be tested.

Reviewer #3: In this interesting paper Pasquero et al follow up previous studies from their laboratory, on HCMV and beta-coronaviruses and from others, showing that viruses can stimulate a type of post-translational modification, deimination or citrullination, of many proteins in infected cells by stimulating peptidyl dearginases (PADs). Here they show Herpes simplex virus specifically stimulates transcription of PADs 2, 4 and especially 3 to enhance citrullination of cellular proteins to the growth advantage of the virus. Specific inhibition of PAD3 by the inhibitors HF4 or CAY10727 or by siRNA markedly inhibited HSV1 replication. Two key HSV inhibitory ISGs, IFITs1 and 2, were also heavily citrullinated, which was not via interferon, and their depletion enhanced HIV replication.

The experiments are generally well done and controlled and key results are highly significant

**Part II – Major Issues: Key Experiments Required for Acceptance**

Reviewer #1: 1. It is very interesting that PAD3 protein levels are undetectable in HFF cells until after HSV-1 infection, and that HSV-1 specifically seems to need PAD3 in this cell type. However, as the authors did not test the specific PAD3 inhibitors in the other cell types included in this study (ie. the epithelial ARPE-19 cells or the SH-SY5Y neuroblastoma cells), or any other cell types for that matter, the conclusion can only be made about HFF cells. It is troublesome that there is no detectable basal levels of PAD3 in HFF cells, which begs the questions what does HSV-1 infection look like in cells with basal levels of PAD3? Clearly HFF cells express basal levels of PAD2 and 4, and HSV-1 infection appears to up regulate these isoforms as well, but with different kinetics and to different levels. Would this happen with PAD3 too, in a cell type that expresses basal levels of PAD3? Is what is observed with PAD3 in HFF cells just a cell-type specific phenomenon? The authors do show that general, pan PAD inhibition inhibits HSV-1 replication in additional cell types, but the conclusions about the PAD3 isoform specifically would be made stronger if the qPCR and westernblot analysis was done in these cell types as well. It would be greatly beneficial if the authors also included a cell type that had basal levels PAD3 expression, like HEK293T cells see citation : (Pong U, et.al. 2014, https://doi.org/10.1016/j.bbamcr.2014.02.018). The same is true for the PAD3 inhibition experiments done with HF4 (lines 184-210), these experiments should be repeated in the ARPE-19 and SH-SYY5 cell types, and perhaps HEK293T cells. These experiments would address if the isoform specificity found with HSV-1 infection is simply a matter of the HFF cells themselves, or something that occurs in all HSV-1/PAD3 relevant cell types.

2. The qRTPCR data presented in Figure 2A and discussed in the text lines 152-156 is normalized to GAPDH, which is an insufficient housekeeping gene to use with HSV-1 infection. It is very well documented that GAPDH is not a sufficient housekeeping gene to use for HSV-1 infection, as this transcript is virtually degraded by the vhs protein by 6hpi. see references (Hsu et.al. 2005, https://doi.org/10.1128/JVI.79.7.4090-4098.2005, and Watson, et.al. 2007 https://doi.org/10.1186/1743-422X-4-130) Due to this reason, the data presented at 16hpi, 24hpi and 32hpi in Figure 2A is likely an artificially inflated mean fold change. Therefore, the experiment presented in figure 2A should be repeated using a different method, such as qRTPCR using a standard curve made from plasmids encoding each PAD isoform, or an RNA-seq experiment where each gene of interest can be addressed individually over time. This is particularly relevant because the data presented in Figure 2A makes it appear that the basal levels of the PAD isoforms are equal in HFF cells, and that the lack of PAD3 protein expression in this cell type is due to regulation after the transcriptional level. This is confusing for the reader, particularly when the authors go on to mechanistically address this phenomenon as stated on line 166, using cycloheximide and IFN gamma experiments, as stated on line 166. Together the cycloheximide and IFN-gamma data only demonstrate that viral protein production is required for PAD3 up regulation, and that this up regulation is independent of the IFN response. This combined with the artificially inflated qRTPCR data make it difficult to understand what is actually going on with PAD3 expression during HSV-1 infection. In order to gain mechanistic insights into PAD3 up-regulation many more experiments need to be included than what the authors have currently done. These experiments would analyze PAD3 expression at the transcriptional level, the protein translational level, the mRNA decay level, and the protein stability level. PAA experiments should also be included to determine if PAD3 up regulation is due to a viral immediate early/ early protein, or a viral late protein.

3. It is well documented that HSV-1 infection causes an influx in intracellular calcium. Seeing that PAD3 expression is induced by an influx of intracellular calcium, it is possible that PAD3 up-regulation is a result of this rather than a direct up regulation of PAD3 gene expression by the HSV-1 viral proteins themselves. HSV-1 infection progresses though the temporal cascade much quicker than HCMV infection in HFF cells, it is possible that the influx of calcium occurs to a much greater level and to a quicker extent than in HCMV infection. This could be an alternate explanation to the results surrounding PAD3 specificity in HSV-1 infection which needs to be explored further. Experiments using artificially induced intracellular calcium levels should be included to rule out this possibility as a mechanism of action—similar to what the authors did for IFN gamma.

Reviewer #2: None noted.

Reviewer #3: However there are some important issues to be addressed:

Figure 1F: The ‘pretreatment’ arm is not really pretreatment as it overlaps with HSV addition for 2 hours. Normally pretreatment means washing off prior to viral addition and if done in that way may provide an even clearer discrepancy in panel G with ‘post -treatment and a better indication of potential mechanism.

Supplementary Fig 1F: the protein citrullination pattern according to molecular weight differs from Figure 1A. Why?

Figure 2c: In Discussion the authors speculate that HSV immediate early proteins are responsible for stimulating the PADs and citrullination, a key question. To differentiate IE/E proteins from late structural proteins the experiment should be repeated with foscarnet (and if confirmed, eventually with multiple IE/E protein mutants).

**Part III – Minor Issues: Editorial and Data Presentation Modifications**

Reviewer #1: 1. Image J should be used to quantify the data presented in the westerblots, particularly surrounding the PAD2 and PAD4 isoforms that do have basal levels in mock cells. This way all western blots, even the ones not shown in the paper could be used the quantification.

2. The westerblots for PAD3 in Figure 2 look underexposed in comparison to the westerblots for PAD2 and PAD4. This is concerning as the authors go to great lengths to point out that there is no PAD3 protein expression in mock infection. The difference in background color on the blots between PAD2, PAD3, and PAD4, particularly in Figure 2E, is questionable.

3. The authors point out that many viral proteins were found to be citrullinated in their screen at 16 and 24 hpi, in addition to the IFIT1 and IFIT2 proteins among others in Figure 4 A and B. The authors use an IP with an anti-peptidylcitrulline antibody to pull down IFIT1 and IFIT2 to show they are citrullinated during infection. As UL54 (ICP27) was shown to be citrullinated at 16hpi in Figure 4B, the authors should be able to pull ICP27 down with the antibody used for the IP in Figure 4C. They should do this to confirm their results in their volcano plot, and it should be easy as they already have all regents required.

4. The authors should add to the discussion after line 315 to speculate why there are citrullinated viral proteins during HSV-1 infection.

5. The way the figures and the supplementary figures are presented is very confusing, and it makes it difficult for the reviewer/reader to find the figures that are being discussed. Many of the figures and the supplementary figures and legends have been separated. For example Figure 1 A, B and C are presented as a separate figure from Figure 1 D, E, F and G. They each have their own legend, and therefore should be renumbered to be their own figures. The authors have done this with supplementary figures as well. The figures and supplementary figures should be re-numbered so each legend describes its own figure and has its own individual number. Or the panels and legends need to be combined for each number. The text for this needs to be altered as well in order to make it more logical.

6. It is unclear if an MOI of 1 was used for every experiment presented in the paper. The methods section needs include if an MOI of 1 was used for the qRTPCR experiments the westerblots etc.

7. The legend for Supplementary figure 2 is missing the correct description of (A). Currently (A) describes (B) and (B) describes (C).

Reviewer #2: 1.With IC50s in the micromolar range in cell culture (61 uM for C1-A), it is difficult to characterize the PAD inhibitors as potent. The authors are requested to tone this down.

2.I don’t think host-directed antiviral therapies have delivered yet on their promise. They have been a popular area of research for more than a decade. There are several examples of drugs that boost the immune response non-specifically or specifically. However, there are disappointingly few antivirals approved for use in humans that target host factors critical for a virus’ replication cycle. Just a comment on the potential of HTAs.

3.The legend of Fig. S2 does not match the panels shown.

Reviewer #3: Supplementary Figure 2 Legend: Legend to panel A is missing so the panel numbering is out of order.

Figure 3: Same heading as figure 2: Specify by siRNA

Supplementary Fig 3C: shows an important confirmation of the role of PAD3 so it should be paired with main figure 3A

Supplementary Figures 4A and B are interesting and part of the narrative and should be main figures

PLOS authors have the option to publish the peer review history of their article (what does this mean?). If published, this will include your full peer review and any attached files.

Reviewer #1: No

Reviewer #2: No

Reviewer #3: No
---

## [Editor Report · Decision Letter 1]

20 Nov 2023

Dear Dr. De Andrea

We are pleased to inform you that your manuscript 'Citrullination profile analysis reveals peptidylarginine deaminase 3 as an HSV-1 target to dampen the activity of candidate antiviral restriction factors' has been provisionally accepted for publication in PLOS Pathogens.

Best regards,

Donna M Neumann

Academic Editor

PLOS Pathogens

Alison McBride

Section Editor

PLOS Pathogens

Kasturi Haldar

Editor-in-Chief

PLOS Pathogens

orcid.org/0000-0001-5065-158X

Michael Malim

Editor-in-Chief

PLOS Pathogens

orcid.org/0000-0002-7699-2064
---

## [Editor Report · Acceptance letter]

30 Nov 2023

Dear Prof. De Andrea,

We are delighted to inform you that your manuscript, "Citrullination profile analysis reveals peptidylarginine deaminase 3 as an HSV-1 target to dampen the activity of candidate antiviral restriction factors," has been formally accepted for publication in PLOS Pathogens.

Best regards,

Michael Malim

Editor-in-Chief

PLOS Pathogens

orcid.org/0000-0002-7699-2064